# Neuron–Glia Interaction in the Developing and Adult Enteric Nervous System

**DOI:** 10.3390/cells10010047

**Published:** 2020-12-31

**Authors:** Verena Pawolski, Mirko H. H. Schmidt

**Affiliations:** Institute of Anatomy, Medical Faculty Carl Gustav Carus, Technische Universität Dresden School of Medicine, 01307 Dresden, Germany; verena.pawolski@tu-dresden.de

**Keywords:** neuron-glia interaction, neural crest cells, enteric nervous system, gastrointestinal system, gut, GDNF, EDN3, Notch, BMP, hedgehog

## Abstract

The enteric nervous system (ENS) constitutes the largest part of the peripheral nervous system. In recent years, ENS development and its neurogenetic capacity in homeostasis and allostasishave gained increasing attention. Developmentally, the neural precursors of the ENS are mainly derived from vagal and sacral neural crest cell portions. Furthermore, Schwann cell precursors, as well as endodermal pancreatic progenitors, participate in ENS formation. Neural precursors enherite three subpopulations: a bipotent neuron-glia, a neuronal-fated and a glial-fated subpopulation. Typically, enteric neural precursors migrate along the entire bowel to the anal end, chemoattracted by glial cell-derived neurotrophic factor (GDNF) and endothelin 3 (EDN3) molecules. During migration, a fraction undergoes differentiation into neurons and glial cells. Differentiation is regulated by bone morphogenetic proteins (BMP), Hedgehog and Notch signalling. The fully formed adult ENS may react to injury and damage with neurogenesis and gliogenesis. Nevertheless, the origin of differentiating cells is currently under debate. Putative candidates are an embryonic-like enteric neural progenitor population, Schwann cell precursors and transdifferentiating glial cells. These cells can be isolated and propagated in culture as adult ENS progenitors and may be used for cell transplantation therapies for treating enteric aganglionosis in Chagas and Hirschsprung’s diseases.

## 1. Introduction

The enteric nervous system (ENS) developed in evolutionary terms independently and earlier than the central nervous system (CNS) [1]. With its 200–600 million neurons in humans, the ENS contains as many neurons as the neural tube and is, therefore, considered the largest part of the peripheral nervous system (PNS) [2]. The ENS is a complex, autonomous nervous system which coordinates all bowel functions [3]. It is connected with the CNS via the sympathetic, parasympathetic and sensory system. Most prominent is its connection to the brain by the vagus nerve for the communication of content composition and hunger [4,5]. In this way, stimulation of the CNS by the gut influences hippocampal learning and memory and regulates motivation and emotions [6,7]. A key component are the gut bacteria. The gut microbiome heavily interacts with the ENS and influences gut physiology [8,9]. The ENS modulates immune cells within the gut wall, which influence the microbiome and the formation of cancer, abdominal pain and neurological disorders such as Parkinson’s disease [10,11,12,13,14,15,16]. The diversity in neuronal and glial subtypes, as well as the diversity of neurotransmitters, is comparably rich as in the CNS [17]. Not all aspects of neuron-glia interaction are fully understood yet, but it is evident that in both systems, physiological neural development and function depend on their functional interaction [18,19]. In the CNS, neurons and glial cells influence each other during maturation and function. Glial cells promote neurogenesis, coordinate neuronal migration, axon growth, the formation of synapses and are involved in neural circuit functions [19]. Glia are themselves regulated by neurons which modulate their transcriptome and numbers, probably to shape the neural networks in the brain [20]. In the ENS, glial cells are similarly crucial for neuroprotection, neural maturation, synapses formation, as well as the release and degradation of neurotransmitters [18,21,22].

The multipotent stem cells of the CNS are the radial glial cells, which may differentiate into glia and neurons [23]. In the ENS, compatible functions are fulfilled by enteric neural crest-derived cells (ENCC), which represent the enteric neural progenitor population [24]. Enteric neurons are distributed along the digestive tract in two different plexuses of ganglia: the submucosal plexus, and the myenteric plexus (Figure 1a). The submucosal plexus consists of a concentric arrangement of enteric ganglia in the submucosa, the tissue layer adjacent to the luminal epithelium. The myenteric plexus is dispersed between the circular and longitudinal smooth muscle layers of the enteric canal. Several signalling pathways are associated with plexus patterning (Sonic hedgehog (SHH), bone morphogenic protein 2 and 4 (BMP2 and 4) Netrin), neural differentiation (Neurotrophin-3, BMP2, 4), gliogenesis (Neuregulin 1 (GGF2), Notch) and axonal growth (Semaphorin 3A, Neuturin) [25]. However, on what stage of differentiation these molecules act and how they interact with each other remains mostly unknown.

## 2. Neurons and Glia during Enteric Nervous System (ENS) Formation

### 2.1. Neural Precursors Arise from Multiple Sources

In recent years, many studies have helped us to understand the molecular mechanism of ENS development [25,26,27]. Developmentally, enteric neural progenitors arise from various sources. Like other neural cells of the PNS enteric neural cells originate from neural crest cells (NCC). The neural crest (NC) is a transient embryonic structure on the dorsal aspect of the closed neural tube. Along the cranial to caudal axis of the neural tube, cranial, vagal, trunk, and sacral NC portions are distinguished [28]. NCCs can differentiate into many different cell types and participate, for example, in the formation of facial cartilage, bone and muscles, septates and the outflow tract of the heart, melanocytes and ganglia of the PNS. In the trunk, NCCs delaminate and migrate ventromedially between the neural tube/notochord and the somites. They first coalesce into dorsal root ganglia and more ventrally into sympathetic ganglia. ENS-fated NCCs mainly derive of the vagal level (commonly defined as region adjacent to somite 1–7) of the NC and in a smaller portion of the sacral level (Figure 1b) [29,30]. Besides ENS development, vagal NCCs also contribute to the pharyngeal arches and the heart outflow tract [31].

Cell labelling experiments in mouse and chicken showed that even the ENS-fated NCCs are not a homogeneous cell population. Cells of different somite levels contribute in varying amounts to the ENS [32,33]. NCCs, adjacent to somite 1 and 2, form Schwann cell precursors (SCP) which become associated with the vagus nerve. Together with the nerve, the SCPs enter the esophagus and the stomach wall where they form parasympathetic ganglia and contribute to about half of the neurons in these two organs [32]. NCCs from somite level 3 to 7 participate in the formation of the superior cervical ganglia, but the majority form enteric neurons and glial cells [34]. NCCs from the vagal level next to somite 3 and 4, contribute in higher numbers to the ENS as compared to other levels. Commonly, vagal NCCs enter the foregut in the esophagus region and migrate as enteric neural crest-derived cells (ENCC) in chains caudally along the growing gut to the anal end [33,35]. The vagal ENCCs cover the longest migratory distance known for any embryonic cell type.

Sacral NCCs contribute to the ENS to a lesser extent. They are derived from the most caudal NC region (caudal to somite 28 in chicken and somite 24 in mouse, post umbilical level). Sacral NCCs migrate ventromedially and, near the distal hindgut, they form the precursors of the prospective pelvic ganglia and invade the rectum and descending colon following extrinsic nerve fibers. There, they become intermingled with caudally migrating vagal ENCCs [36,37]. While migrating, some ENCCs start to differentiate [38,39,40]. Differentiation continues during the whole embryonic and fetal period and up to three weeks after birth [41,42].

Interestingly, vagal and sacral ENCCs seem to be two entirely different cell populations. Not only their place of origin, the timing of induction and contribution to the ENS differ, but also their migration is counter-directed—vagal ENCCs migrate in an oral to anal direction, and sacral ENCCs do so vice versa. Also, vagal ENCCs are more invasive than sacral ones as they colonise the entire gut. This migratory behavior is cell-intrinsic. Vagal ENCCs migrate further than sacral ENCCs even when they are grafted into a sacral NC environment [43]. Eventually, both populations form histologically and functionally identical neurons and glial cells.

The ENS might have even a more mixed origin. Lineage tracing experiments with *Pdx1-Cre x Rosa-YFP* mice identified several pancreas-derived neurons in the whole bowel. *Pdx1* expressing pancreas progenitors participate in the formation of enteric neurons but not in glial cells formation [44]. In culture, *Pdx1*-expressing cells form neurospheres, indicating their self-renewing potency. Conversely, vagal ENCCs are involved in pancreas development where they form neurons and Schwann cells of intrinsic ganglia [45]. The pancreas develops from the endodermal lining of the foregut. The initial pancreas bud serves as an entry site for NCCs [46]. In total, both the gut and the pancreas supply each other with neural progenitors and thereby increase the diversity of neuronal origin in the gastrointestinal system.

Furthermore, in the mucosal region of the small intestine of *Ret*-deficient mice, which do not form an ENS, neurons associated with extrinsic nerve fibers were identified [47]. These neurons are derived of SCPs. They develop from trunk NCCs and undergo neurogenesis independently of the ENCCs (Figure 1b). The SCPs start to differentiate and to express neural markers approximately 21 days after birth. In comparison, vagal ENCCs begin to form neurons at E10.5 [41]. In young adult mice, SCP-derived neurons contribute to 5% of submucosal ganglia in the small intestine and to 20% of neurons in the submucosal and myenteric ganglia of the large intestine [47]. Mostly, these neurons are excitatory motor neurons, interneurons and intrinsic primary afferent neurons. Specific elimination of the SCP-derived neurons resulted in a decrease of the overall enteric neuron numbers in adult mice. Therefore, it seems that ENS formation depends on different sources of neural progenitors to form neurons and glia in the whole gut. So far, it is not known how the diversity of the precursor populations is regulated and how individual cell numbers are balanced.

Irrespective of their origin, ENCCs form a heterogeneous cell population. Lineage tracing experiments and clonal analysis using a *Sox10: confetti* mouse line identified three distinct groups of ENCC subtypes (Figure 1c): proliferating bipotent neuron-glia progenitors cells as well as neuronal- or glia-fated ones [48]. At E12.5, cells expressing the ENCC marker *Sox10* were randomly fluorescently labeled by the expression of either *Gfp* (green), *Rfp* (red), *Yfp* (yellow) or *Cfp* (cyan) upon tamoxifen administration. By that, all clonal descendants of these initially *Sox10* expressing ENCCs were traced later. The ratio of neural- or glial cell-fated individual clones provides information about the previous potency of the precursor. With this approach, cells with a former neuron-glia bipotency, neuronal fate and glial fate were identified [49]. The majority of clonal groups were composed exclusively of glial cells. About a quarter of the clonal groups had a mixed neuron–glial identity, and only a fraction of clones displayed an exclusive neuronal identity. This population were also smaller in cell number as compared to the other two groups. Therefore, neuronal precursors seem to have a limited proliferation capacity. To keep up progenitor identity, ENCCs need to express *Sox10*, which maintains bipotentiality and inhibits neural differentiation [50]. ENCCs colonise the gut wall in an outside to inside manner [49]. First, ENCCs migrate within the region of the later myenteric plexus between the muscle layers. Neuronal-fated progenitors differentiate only in this layer into neurons. In a second colonization wave, glial-fated and bipotent ENCCs may migrate into the luminal region of the prospective submucosal plexus where they differentiate further into neurons or glia respectively. Furthermore, glial cells can migrate into the mucosal layer where they are known to form units with blood vessels, epithelial cells and immune cells [10,49,51]. Thus, the distribution of neural precursors along the entire bowel, the migration into the specific tissue layers is a highly coordinated process.

### 2.2. Enteric Neurons

In the small and large intestine, neurons differentiate before glia [52]. Upon neurogenesis, neuronal-fated progenitors maintain the expression of *Ret* and *Phox2B* but additionally express the neuron-specific genes *Tubb3*, *Nf*, *TH*, *PGP9.5* and *Elavl4*. As they differentiate, they become negative for glial markers and do not further proliferate [49,52].

Precursors of different neuronal cell types exit the cell cycle at different time points during embryonic development [41]. In a classical study, pregnant mice or pups were injected with [3H]thymidine at different time points over the whole ENS formation period. Neural subtypes were immunohistochemically identified 30 days after birth by the incorporation of [3H]thymidine, and specific neuron markers [41]. The earlier formed neurons produced serotonin (5-HT) and differentiated between E8-E14 (peak at E10), followed by choline acetyltransferase (ChAT) positive neurons (E8-E15, peak at E12), enkephalin (E10–E18, peak at E14) and neuropeptide (E10–E18, peak at E15) producing neuronal subtypes (Figure 2). Later, vasoactive intestinal peptide (VIP)/NADPH positive (E10–P5, peak at E15) and calcitonin gene-related peptide positive (CGRP) (E10–P3, peak at E17) neurons were formed.

Two single-cell transcription analyses identified nine molecularly distinct enteric neuronal subsets in the myenteric plexus of the small intestine, 24 subgroups in the colon of mice and 11 in humans [53,54]. In both studies, neurons were grouped broadly into either cholinergic (ChAT-positive) or nitrergic (nitric oxide synthase (NOS)-positive) subgroups. The expression profile of neurotransmitter receptors, serotonin, acetylcholine and the potassium and sodium channels varied in neurons located in different parts of the colon. Therefore, the location is beside the circadian clock influential on neural gene expression, further increasing the complexity of the ENS functional regulation [54].

All these neural subtypes form a network of interconnected intrinsic sensory neurons which receive signals from the epithelium and muscle layers about nutrition content and mechanical distortion. The signal is passed via synaptic connections over descending and ascending interneurons. The final targets are excitatory muscle motor neurons and intrinsic primary afferent neurons, which modulate gut motility [17,55]; see also for a detailed review on functional circuits and signal transduction].

An additional layer of signal control is mediated by interstitial cells of Cajal (ICC) and telocytes (TC), also referred to as fibroblast-like cells positive for the platelet-derived growth factor receptor α (PDGFRα+ cells) [3,56,57,58]. ICCs and TCs have a mesenchymal origin and TCs could differentiate into ICC upon damage [58,59,60]. Both cell types are found in several organs, including the gut, where they formed a network between and within the muscular layers. TCs are characterized by long telopods which are in contact with blood vessels, smooth muscle cells (SMC), nerves, macrophages and other immune cells [57,61,62]. TCs are believed to give mechanical support on a cellular level and facilitate cell-cell communication over cell contacts. In the intestinal crypts, subepithelial TCs support epithelial renewal and stem cell proliferation by secreting Wnt proteins. Additionally, they regulate epithelial gene expression in intestinal villi [63,64,65]. ICCs modulate electrical signal transduction from motorneurons to SMCs by generating a slow electrical wave which transduces phasic contraction of SMCs [3,56,57]. Cellular contact and gap junctions were identified between TCs and ICCs as well as ICCs and SMCs [58]. TCs might act as postjunctional cells involved in purinergic neurotransmission [62]. In the myenteric ganglia of the ileum, TCs formed a continuous layer around ganglia, whereas the ICCs were individually located between TCs and circular SMCs [66]. By contrast, telopods were found inside ganglia in the colon. The functional coupling of SMCs, ICCs and TCs is necessary for organized peristaltic movements and defined them as a functional “SIP syncytium” (naming after SMCs, ICCs and PDGFRα^+^ cells) [56].

### 2.3. Enteric Glial Cells

The enteric glial cells start to form from E12 onwards and can show a high variation in morphology and functionality [67,68]. Enteric glial cells continue to mature postnatally up to 4 weeks after birth, in young rats [69]. *Sox10*, *Phox2B*, *Ret*, *p75^NTR^*, *Erbb3* and *Fabp7* are markers for the undifferentiated ENCCs [49,52,67,70]. Glial-fated progenitors start to express the gene of the proteolipid protein 1 (*Plp1*) and continue the expression of the progenitor genes *Sox10*, *Erbb3* and *Fabp7*. Differentiating glial cells downregulate *Ret*, which has a pivotal role in neural differentiation.

Distinct subtypes were identified based on their morphology, and the presence of glial markers such as the glial fibrillary acidic protein (GFAP), the Ca^2+^-binding protein S100β and the transcription factor SOX10. Intraganglionic glial cells often show multiple irregular and branched processes which connect with various neurons within the ganglion. These cells are grouped as astrocyte-like, “protoplasmic” or type-I glial cells [68,71]. The type-II glia are fibrous with processes parallel to neuronal fibers which get in contact with the neurons but do not wrap around them like Schwann cells. Type-II glia are typically located at the edges of the interganglionic connectives or lie outside the ganglia. Often, they show four branched processes and tend to come into contact with small blood vessels. Supposedly, they form a matrix suitable for the neuronal fibers outside the ganglia [68,72]. Glia with a similar morphology are present within the lamina propria of the mucosa [68,73,74]. Type-IV glial cells have a characteristic bipolar morphology and are located along nerve fibres within the circular and longitudinal smooth muscle layers [68,74,75]. GFAP is only present in a subset of enteric glial cells [70,76]. This marker is strongly expressed in type-I glia but only in a few type-II and type-III glial cells [68]. Temporally, the expression may vary. Glial cells which produce GFAP at one time point might not exhibit it seven days later. Currently, it is unclear whether GFAP presence is a result of mechanical forces, physiological functions or a marker of mature glia. A comparative immunohistochemical analysis of the adult mouse myenteric plexus showed that all glial subtypes expressed either one or two of the glial marker GFAP, S100β or SOX10 [68]. Nearly all cells which produced S100β were additionally positive for SOX10 but not vice versa. These variations are more likely due to dynamic gene regulations than real linage restrictions. All of these factors are functional to mark enteric glial cells, but none is sufficient to label all glia. The majority of glia expresses *Plp1*, *Sox10* or *S100β* [70].

The gut microbiota controls glial development and homeostatic renewal throughout the adult life. Consequently, antibiotic treatments impaired glial homeostasis [77,78]. The exact mechanism remains enigmatic, but one factor that links the microflora and the ENS are macrophages. Macrophages have a haematopoetic origin and colonise the embryonal gut independently of the ENS [79]. Distinct types of macrophage exhibit different transcriptomes, linked to specific functions and locations [80]. Mucosal macrophages modulated gut homeostasis and secretion by the interaction with neurons and blood vessels. In contrast, myenteric macrophages influenced ENS formation directly via the secretion of BMP (bone morphogenic proteins) molecules [81,82]. In turn, enteric glial cells activated intraganglionic macrophages via connexin 43 channels and the secretion of macrophage colony-stimulating factor (CSF) [83,84]. Further, they regulated group 3 innate lymphoid cells and thereby orchestrated gut defence [10]. Commonly, enteric glial cells are part of a tight regulatory circuit between the microbiota and the immune system [3,85,86,87].

Based on their developmental origin as NCC derivatives, enteric glial cells and Schwann cells were initially believed to share a close relationship. Morphologically, type-I enteric glia share similarities with astrocytes. However, a comparison based on RNA sequencing revealed that enteric glia have a unique transcriptome as compared to other glial types [70]. They express markers of all glial cell types and even share some transcripts with neurons. The enteric glia express some Schwann cells specific genes like *Sox10*, *Plp1*, *Mbp* and *Mpz*, but also the astrocyte markers *Gfap*, *Entpd2* and *Dio2*.

### 2.4. Molecular Control of Neural Precursors Migration and Differentiation

#### 2.4.1. Glial Cell-Derived Neurotrophic Factor (GDNF)/RET Signalling

The migration of NCCs into the gut and to the anal end is guided mainly by two signalling systems: glial cell-derived neurotrophic factor (GDNF)/RET and endothelin 3/endothelin receptor type B (EDN3/EDNRB) [25,26,88]. Both signalling pathways were studied thoughtfully over the last decades and are the content of several reviews [24,88,89,90,91,92,93,94]. ENCC express the receptors RET, a transmembrane tyrosine kinase, whereas the ligand GDNF (glial cell-derived neurotrophic factor) is secreted by mesodermal cells of the surrounding gut mesenchyme [95]. Additionally, GDNF is secreted by glial cells, intestinal smooth muscles and epithelial cells [18,96,97,98]. Mice with a loss of function of either GDNF, RET or its co-receptor GFRα (glycosylphosphatidylinositol (GPI)—anchored co-receptor GDNF family receptor α1) fail to develop an ENS [99,100,101,102,103]. *Gdnf* is initially expressed in the stomach at E9.5 when vagal NC migration starts and is present in all gut regions at E11 in mice. In explants and in vitro, GDNF reliably attracted ENCC and overexpression or systemic administration of GDNF lead to an increase in ENCC numbers (Figure 3) [42,95,104]. In sum, GDNF/RET signalling promotes ENCC survival, proliferation and migration [49]. GDNF supports neurogenesis of neuronal-fated ENCCs but does not induce neural fate [42]. Local overexpression in glia, as well as systemic administration of GDNF, enhanced the differentiation into the timed neuronal subtype. However, it did not increase the overall neuron number, nor the amount of individual neuronal subtypes whose formation was either not started or was already terminated at the time of the experiment. Elevated excess of GDNF in the *Gfap-Gdnf* mouse model from E17 on, lead to an increase in differentiation of neural progenitors into NADPH-positive neurons but not into the earlier arising ChAT-positive neurons [42].

#### 2.4.2. Endothelin 3/Endothelin Receptor Type B (EDN3/EDNRB) Signalling

EDN3/EDNRB promotes GDNF/RET signalling [105] and supports the maintenance of the neural progenitor state (Figure 3) [92,106,107]. Gut mesenchymal cells secrete EDN3 which binds to EDNRB on the ENCC surface. Binding of the receptor leads to a G protein-mediated activation of phosphoinositide 3-kinases (PI3K), as well as the cAMP/cAMP response element-binding protein (CREB), phospholipase C (PLC) and Rho GTPase pathways. The activation of these pathways promoted the modulation of the cytoskeleton, NO-production and Ca^2+^-release. As a long-term effect, EDN3 promoted ENCC proliferation and migration but inhibited neural differentiation in vitro and in vivo [92,107].

Besides the secretion of chemoattractive molecules, the gut mesenchyme secrets an extracellular matrix (ECM) suitable as a substrate for migrating ENCC. EDN3/EDNRB signalling enhanced the migration of ENCCs by stimulation of cell adhesion to ECM components via β 1-integrins [108]. A β1-integrin knock-out mouse model displayed slow ENCC migration behavior in the small intestine and no invasion of the hindgut, but unaltered ENCC proliferation rates [109]. The ENCCs themselves are also capable of influencing their microenvironment by the secretion of collagen 18 and agrin [110]. Collagen 18 supports ENCC migration, while agrin limits cell movement and surrounds differentiated glial cells and neurons. The gut tissue accumulates laminin, which stiffens the ECM and, thereby, slows ENCC migration over time [111]. Furthermore, laminin also determines the invasion capacities of ENCCs in vivo and their differentiation in vitro [112,113]. This indicates that there is a limited time frame for the ENCCs to colonise the gut successfully.

#### 2.4.3. Notch Signalling

Notch signalling has a dual role in ENS development. First, it maintains the ENCC progenitor state by the initiation of *Sox10* expression. SOX10 and ZEB2 are transcription factors characteristic for ENCCs, and both directly activate the *EdnrB* promoter [114]. The receptors Notch1 and Notch2 and the ligands delta-like 1 (DLL1), and delta-like 3 (DLL3) are the main participants of the Notch pathway in the ENS [115]. Indirectly, the Notch target gene *Hes1* supports *Sox10* expression by inhibition of *Mash1*. MASH1 promotes neurogenesis by repression of *Sox10.* Besides the maintenance of the undifferentiated ENCC progenitor state, SOX10 is necessary for glial identity [50]. That leads to the secondary function of Notch: the promotion of gliogenesis (Figure 3).

An ENCC specific downregulation of *Pofut1*, a modifier of Notch pathway, resulted in a reduced number of ENCCs by forcing cells to leave their progenitor state. Additionally, premature differentiation and an increase of neuron numbers were observed at the expense of glial cell numbers [115]. That was caused by the downregulation of *Sox10* and upregulation of *Mash1* in Notch deficient mice. If SOX10 was reduced and *Ednrb* expression missing differentiating neurons stopped migration, exited the cell cycle and started to differentiate [116]. The premature exit of the cell cycle and onset of differentiation leads to inadequate colonisation of the gut by depletion of the ENCC pool [115,117]. A similar role of Notch signalling in the maintenance of neural stem cell identity and gliogenesis was previously reported in other parts of the PNS and CNS, pointing towards a conserved function of Notch in neural development [118,119,120]. However, the regulation of the Notch pathway in the ENS might very well be different. In the gastrointestinal tract, epithelial hedgehog signalling (Hh: indian hedgehog (Ihh) and sonic hedgehog (Shh)) balances Notch in ENCCs and the mucosal mesenchyme via the amount of available DLL1 [117,121]. Its right amount is crucial as an under- or oversupply of Notch in the gut mesenchyme both lead to an inadequate ENCC colonization and premature gliogenesis. An absence of Hh signalling leads to an oversupply of Notch and overexpression of Notch phenocopies Hh-deficiency. Therefore, Hh signalling is considered as a direct upstream regulator of Notch signalling in the gut (Figure 3) [121].

Furthermore, Shh signalling may regulate Notch indirectly as well by modulating the composition of the ECM by upregulation of the synthesis of collagen I, collagen IX, chondroitin sulfate proteoglycan (GSPG) and versican [122]. An excess of collagen IX and versican inhibited ENCC migration and, thereby, disturbed ENS formation. Notch receptors and ligands share their EGF motives with ECM proteins and bind to ECM components [123]. Hypothetically, Notch signalling in ENCCs may be regulated by the composition of the ECM, or mesenchymal cells may change ECM composition after Notch or Shh activation. Either way, the impact of Hh signalling via Notch on ENS development is very likely.

#### 2.4.4. Bone Morphogenetic Proteins (BMP) Signalling

Bone morphogenetic proteins (BMP) are involved in cell cycle exit of ENCCs and thereby govern neural differentiation (Figure 3) [124]. At E12, they induce neurogenesis by enhancing GDNF/RET signalling and suppressing GGF2-mediated gliogenesis [67]. Overexpression of the BMP-antagonist Noggin in enteric neurons leads to an increase of cell numbers of neuronal subtypes which exited the cell cycle early on the expense of subtypes which withdraw from the cell cycle at a later stage. These earlier forming neurons are marked by 5-HT, calbindin, calretinin and NOS. Especially, the number of serotonergic neurons became more in the myenteric plexus as well as calbindin-positive neurons in the submucosal plexus. At the same time, the portion of gamma-aminobutyric acid (GABA)-, CGRP- and tropomyosin receptorkinase C (TrkC)-positive neurons were reduced in both plexi. These results lead to the hypothesis that BMP signalling has a role in plexus pattern formation. Overexpression of BMP4 increased the number of TrkC/neurotrophin 3 (NT3)-dependent neurons but reduced the number of 5-HT ones [124].

Furthermore, BMP-signalling (BMP2, 4) promoted gliogenesis at E16. It enhanced the expression of the neuregulin receptor *Erbb3* and its ligand *Ggf2* and suppressed GDNF/RET signalling in glial-fated ENCCs [67]. A disturbance of BMP-signalling changed the neuron-glia ratio in favor of neurons. Thus, BMP might be more critical for gliogenesis than neurogenesis.

## 3. Neurons and Glial Cells in the Adult ENS

Postnatally, the ENS matures in allegiance with the establishment of gut functionality and motility [125]. This includes the generation of synapses between glia and neurons. In newborn P1 rats (postnatal day 1), only a few synaptic contacts, as marked by the presynaptic marker synapsin I, were identified. From P7 onwards, robust synapsin I labelling in glia was observed, which was paralleled by an increase in GFAP (marking glia maturation) and further increased in guts of P21 and P36 rats [18]. This might imply that glia maturation is linked to neuronal connectivity in the ENS. This finding is consistent with the response of electric field stimulation, between P7 and P14 in enteric colon myenteric neurons of newborn rats [126]. Coordinated matured colon mobility was detected from P10 on.

The ENS is exposed to mechanical and chemical stress and damage. Therefore, some regeneration mechanism seems evolutionarily implemented. Several partly conflicting reports tried to identify the precise nature of this mechanism. Some indicate that the gut harbours a stem cell population. Approximately 1% of isolated gut cells form neurosphere-like bodies in culture and show an extensive self-renewing capacity (Figure 4a) [127,128,129,130]. These multipotent progenitors are *p75^NTR^*—and *Nestin*-positive and were isolated from fetal mice and rats up to adult stages (P22) [127,128]. The efficiency in generating multiple subclones with neuronal, glial and myofibroblast identity decreased with age. In progenitors from older individuals, mostly glial clones were formed [127]. Furthermore, the total number of clones and the proliferation rate of isolated stem cells decreased (P22 as compared to fetal stage E14) [127,128,130]. It is still controversial as to whether or not these cells are true ENCC-like stem cells or only display a stem cell phenotype if isolated and seeded in a host gut or are grown in culture. If these cells contribute to neurogenesis under normal conditions in vivo remains enigmatic. Joseph and colleagues found no adult neurogenesis during homeostasis, ageing, pregnancy, diet changes, hyperglycemia, exercises, inflammation, irradiation, neurotoxicity, partial gut stenosis and glia ablation [130]. Instead, they reported gliogenesis in homeostasis and allostasis (Figure 4b).

In contrast, Kulkarni and colleagues reported neuronal apoptosis and the occurrence of physiological neurogenesis in the adult mouse intestine [131]. *NOS*-expressing neurons were genetically labelled by the expression of tdTomato red fluorescent protein after tamoxifen injection (*NOS1-creER^T2^*:*dtTomato* mouse line). Seven days later, new neurons were found lacking the tdTomato protein, indicating the formation of new neurons. Apoptotic neurons were identified by *Caspase 9* expression at a rate of 4–5% cell loss per day. A *Nestin/p75^NTR^* coexpressing subpopulation which did not express neural markers was suspected of having progenitor quality and of replacing lost neurons (Figure 4c). Therefore, *Nestin* expressing cells were followed in vivo and found to form neurons in the adult myenteric ganglia [131]. According to their results, enteric neurons are replaced every two weeks. Despite opposing results, both studies found similar glial typic markers expressed by their enteric neural precursor cells: *p75^NTR^*, *Gfap*, *S100β*, and *Sox10* [130] vs. *Nestin*, *p75^NTR^*, partially *Gfap* or *S100β* but not *Sox10* [131]. Others reported adult neurogenesis to occur in vivo only upon stimulation of the 5-HT receptor and chemical or mechanical injury [132,133,134,135,136]. It remains to be investigated if there exists a difference between normal healthy homeostasis and disease or if it is due to the experimental set-up.

Besides resident adult ENCC-like progenitors or stem cells, one potential source for adult neurogenesis could be the SCPs (Figure 4d). Developmentally, they are multipotent and form among other cell types enteric and parasympathetic neurons [137,138]. Generally, ENS formation from NCCs is conserved in higher vertebrates. In zebrafish, new neurons are generated exclusively by SCPs during growth or after injury [136]. In evolutionary terms, SCPs might be the primary source for enteric neurons [139]. A study in the jawless vertebrate sea lamprey (*Petromyzon marinus*) revealed that in lower vertebrates the vagal NCC population is absent, and the entire ENS is formed by SCPs derived of trunk NCCs [139].

Often, simple model organisms are helpful to study developmental or evolutionarily conserved processes. Besides in mice, many studies in the ENS were performed on chicken embryos which offer embryonic manipulations *in ovo*. External developing aquatic animals like fish, amphibians or the sea lamprey are directly accessible, and with their simple body organization, are easy to observe and analyze. For example, the straight and short gut of the sea lamprey or the axolotl embryo (*Ambystoma mexicanum*), makes it easy to follow ENCC migration and ENS formation in vivo.

Other potential sources for adult neural progenitors are the enteric glial cells (Figure 4b), which may transdifferentiate into neurons upon injury or in culture, comparable to the glia of the CNS [76,113,130]. Mice induced by dextran sodium sulfate or *Citrobacter rodentium* induced colitis displayed *Sox2* and *Plp1*-expressing glia directly transdifferentiating into neurons. They did not dedifferentiate or enter the cell cycle again beforehand [76]. Likewise, in patients with colitis induced by *Clostridium difficile* or ulcerative colitis, on average, 14% of newly formed neurons are derived from glia in response to the infection [76]. Consistently, an induction of colitis in vitro increased the amount of neurons [135]. There, an increase in glial cell proliferation in the absence of an accompanying increase in the amount of glial cell numbers was reported. This lead to the hypothesis that enteric glial cells are a source of newly generated neurons.

Initially, cultivated enteric neural cells express the glia marker *Gfap*, and only a fraction expressed the neuronal marker *βIII tubulin* or both markers. After six days, most isolated cells coexpressed *Gfap* and *βIII tubulin* [113]. In vivo, glia markers were never reported to colocalise with neuronal markers. This might be due to environmental cues and the influence of surrounding muscle, connective tissue and immune cells. Accordingly, enteric glial cells cultivated on a 3T3 mouse embryonic fibroblast feeder layer were limited in their potential to differentiate into neurons [113]. In particular, laminin, a prominent protein of the ECM and secreted by fibroblasts, inhibited glial transdifferentiation. In a neuronal cell culture system with indirect co-culture of glia, secreted molecules promoted the formation of synaptic connections between neurons. This effect is mediated by molecules of the purinergic P2Y_1_ receptor- and GDNF-pathway [18]. However, glia have a limited effect on neural differentiation in vitro. In the co-culture system, the number of ganglia and the number of neurons per ganglion is not influenced by the presence of glia.

Previously, it has been concluded from a series of preceding studies, that glial cells have an essential role in maintaining gut epithelial function. Enteric glia were eliminated by the generation of transgenic mice which harbour the thymidine kinase gene of the *Herpes simplex* virus under the control of the *Gfap* promoter [73,140]. The antiviral agent ganciclovir (GCV) was injected into these mice. Glial cells metabolised the GCV into a cytotoxin by the thymidine kinase, resulting in glial cell death. After GCV administration, transgenic mice showed an increased gut epithelial permeability and crypt cell proliferation in the jejunum, accompanied by bowel inflammation [141]. As we know now, not all enteric glial cells express *Gfap* and its expression changes over time [68,70,76]. Therefore, these classic experiments have been repeated. Now, the expression of the diphtheria toxin subunit A gene was controlled by the *Plp1*-promotor to induce glial cell death in the entire population [141]. In this approach, glial cells were broadly eliminated, but no inflammation was observed. Epithelial histology and barrier permeability remained healthy, and no microbial translocation through the barrier could be reported. In female, but not in male mice, gut motility was impaired. In contrast to the *Plp1^CreER^*; *Rosa26^DTA^* model, the *Gfap^HSV-TK^* model showed a rare expression of *Gfap* in epithelial cells and diffusion of toxins into neighbouring cells. Together, this explains the previously reported epithelial barrier damage [141].

## 4. Diseases of the ENS and Therapeutic Approaches

Misregulation of the ENS may cause a variety of severe enteric neuropathies including oesophagal achalasia and gastroparesis [17]. A most extreme form of enteric nervous system malfunction is aganglionosis, the absence of the enteric nervous system in parts of the gastrointestinal system, which occurs in 20–30% of patients with chronic Chagas disease or as a congenital disability leading to Hirschsprung disease (HSCR) [26,142]. In Chagas disease, the infection with *Trypanosoma cruzi* leads to a degeneration of the central and enteric nervous system, whereas in HSCR the ENS formation is impaired prenatally. Agangliosis leads to missing peristaltic movements and permanent contractions of the uninnervated smooth muscles. As a consequence, dilatation of the colon (megacolon) or any part of the gastrointestinal tract may manifest in HSCR or Chagas disease, respectively. In HSCR patients, the ENS is absent in the terminal hindgut to a variable extent [93]. HSCR occurs in 1 in 5000 births and 12% of children with Trisomy 21. Megacolon is associated with variable symptoms. The most serious ones are bowel inflammation, enterocolitis, and perforation. To date, the only therapy for the affected children is the surgical removal of the entire colon or parts of it. Often, life-long complications like enterocolitis, obstipation or incontinence occur.

Commonly, HSCR is characterised by an impairment of migration, proliferation, or differentiation of ENCCs with the involvement of many gene products [143]. About 50% of inherited and 20% of sporadic cases of HSCR show a mutation in the *RET* gene [144,145]. Approximately 5% of HSCR patients show an *Ednrb* gene defect and also mice homozygous for either *Ednrb* or *Edn3* have an aganglionosis of the distal colon [146,147,148].

Cell replacement therapy by transplantation of neuronal precursors into the aganglionic part of the gut is commonly suggested for the treatment of HSCR and Chagas disease [93,149,150]. Experimentally, isolated enteric neural cells from adult mouse and human gut tissue and human embryonic stem cells can form neurospheres in culture which raised the question of whether or not these cells may be applied for cell replacement therapy approaches [44,129,151]. In such procedures, competent enteric neural precursors are propagated in culture and are allowed to differentiate into definitive enteric progenitors. Subsequently, these cells are injected into the bowel of patients to repopulate the aganglionic part of the bowel and form a functional ENS. So far, appropriate neural precursors have been isolated from human myenteric and mucosal biopsies or produced from human embryonic stem cells [129,151,152,153,154]. These cells were propagated as neurosphere-like bodies and were differentiated into GFAP-positive glia as well as 5-HT-, ChAT-, GABA-, NOS-, VIP-, Substance P- and CGRP-positive neuronal subtypes either in culture or upon transplantation into an aganglionic gut of mice or chicken [129,151,152,153,155,156].

Cellular manipulations in culture before transplantation might be necessary to overcome possible intrinsic migration or proliferation deficiencies of the patient-derived material and to enhance the cellular colonisation potential in vivo. However, postnatal ENS progenitors might be less invasive to the gut tissue than they were during the embryonic period. A recent study in a chicken-quail model indicated that ENS precursor cells postnatally lost their capacity to form the ENS [157]. Isolated ENCCs were able to colonise an aganglionic gut, but only if derived from young donors. As soon as the ENCCs were differentiated and incorporated into the neural network, they did not efficiently colonise the aganglionic gut anymore. In mice, transplanted ENS progenitors could form a functional ENS successfully in the myenteric plexus region close to the injection site. However, they did not colonise the entire bowel, and they did not migrate into submucosal areas [156]. Given that this occurs in humans as well, it would compromise the therapy success. Therefore, culture conditions have to be further explored and improved. The amount of mouse-derived ENS progenitors has been increased in culture by exposing the cells to GDNF [158]. These cells showed a 2-fold enhanced migration potential if grafted into the aganglionic colon of mice. Activation of the canonical Wnt pathway in mouse and human ENS neurospheres lead to an increase in neuronal differentiation [159]. Currently, the mechanism of the formation of a functional ENS network from transplanted cells in the aganglionic gut is wholly unknown. A deeper understanding on the control mechanisms of embryonic and adult neural differentiation for functional ENS formation is required.

Advances in the generation of human intestinal organoids and enteroids may facilitate novel research on gastrointestinal diseases [160,161,162]. Initially, organoids were limited to epithelial structures but in combination with NCC derived from human pluripotent stem cells, organoids with a functional enteric neural network can be formed, enabling research on the ENS and associated diseases [163,164]. This opens new possibilities to test neural interactions and allows easy genetic and chemical manipulations. Organoids can be studied in culture or grafted into hosts which further amplifies possible scientific questions and potential therapeutics.

Although transplanted ENS progenitors are intrinsically able to migrate and differentiate, the microenvironment of the gut of affected patients might be inhibitory. A repopulation with transplanted ENS progenitors remains challenging unless the microenvironmental conditions can be improved. An example illustrating this problem is the abundance of collagen VI in the tissue environment of postnatal ENCCs in HSCR patients with trisomy 21. Collagen VI provides only a poor ECM substrate for ENCCs. Therefore, migration is delayed and limited, leading to aganglionosis of the terminal areas of the gut [165]. For a successful restoration of the ENS function in children with HSCR, new therapies and drugs must be developed that guarantee not only the presence of functional neurons but also allow a permissive or supportive cellular and extracellular tissue environment [152]. However, many more preconditions have to be fulfilled for a successful approach, e.g., patient-derived ENS progenitors need be produced in a sufficiently high number, potential difficulties during migrating into all gut tissue layers have to be overcome, differentiation needs to be regulated, and long-term safety and functional restoration of gut properties have to be secured [166]. Nevertheless, the success of the last decades offers hope that these obstacles will be overcome in time.

The first promising results have been obtained by the application of GDNF to HSCR mouse models [167]. As a consequence, survival rates of diseased animals improved, and even neurogenesis initiated by Schwann cells and remaining sacral ENCCs occurred. Furthermore, GDNF improved epithelial barrier function and inflammatory responses in mouse models for ulcerative colitis [168]. Mucosal glial cells secrete GDNF after activation of the toll-like receptor 2 by the gut microbiome [169,170]. Enterocytes are the second source of GDNF but can sense GDNF signalling via the RET receptor. An activation of GDNF/RET signalling facilitated the formation of tight junctions and induced proliferation of enterocytes [98,169]. Thereby, GDNF supports homeostasis and wound healing of the gut epithelium.

Interactions of the gut microbiome and the ENS via the immune system is mandatory for normal functions of intrinsic and extrinsic nerves and gut–brain communication in homeostasis and allostasis [55,87,170,171,172,173]. Direct bacterial–neuronal interaction was mediated by 5-HT, calbindin and NO [9,16,171,172,173]. In turn, macrophages and mast cells interacted with enteric neurons via CSF and BMP 2 [87]. Novel drug treatments in combination with the restoration of the microflora or cell transplantation may lead to therapeutic success in a variety of neuronal and inflammatory diseases of the gastrointestinal tract.

## 5. Conclusions

Research over past centuries unravelled the fundamental processes of developing and adult ENS formation as well as their interaction with the gut microbiome and immune system. This knowledge helps us to understand inflammatory diseases and multifactorial enteric neuropathies like HSCR, and Chagas in order to develop and improve new therapies.

Currently, we still lack a detailed understanding of the interaction of the main signalling pathways GDNF, EDN3, BMP, Shh and Notch. Together, these factors control ENCC proliferation, maintenance, migration and differentiation. Furthermore, the composition of the ECM is highly influential on ENCC migration and most likely plays a role in disease progression. So far, we are just at the beginning of unravelling the role of ECM proteins in ENS formation.

Currently, experimental therapies focus on providing new NC material to patients. The gut microenvironment is mostly uncharacterized and might not be supportive of neural cell survival. Therefore, in order to improve cell transplantation therapies, the ECM composition needs to be understood and modified. One possible way is the stimulation of gut mesenchymal fibroblasts, as the primary source of ECM proteins, with BMP or Notch. In this way, an appropriate embryonic-like neural supportive matrix composition may be presented to embryonic-like neural progenitors which are transplanted into the bowel of patients.

Additionally, the determination of neuronal or glial cell fate is unknown. This knowledge is crucial to control the composition of transplanted neural cell types. Neural precursors propagated in vitro colonize the gut and also partly restore gut function but the underlying mechanisms remain enigmatic. Currently, it is still unclear whether transplanted cells form a new ENS or if they initiate intrinsic regeneration processes. Additionally, the role of stem cells, glia and SCPs for ENS regeneration in gastrointestinal diseases needs to be unravelled. These are exciting topics for future studies as research on the ENS over recent years has offered a promising path for the treatment of ENS-associated diseases.

## Figures and Tables

**Figure 1 cells-10-00047-f001:**
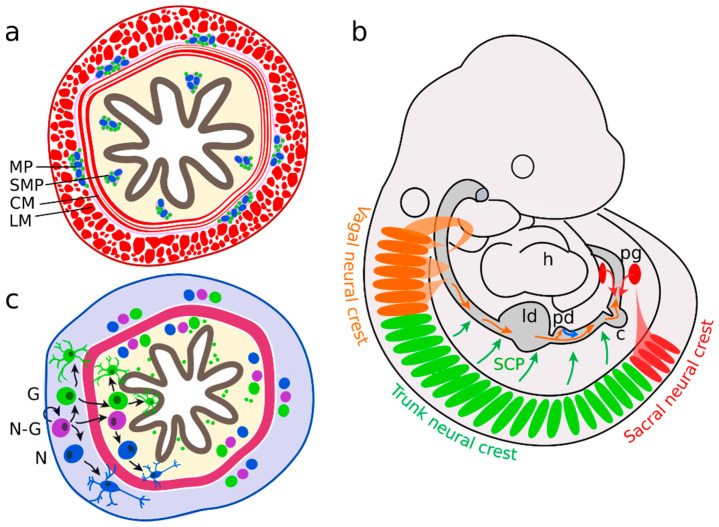
Enteric nervous system (ENS) precursors colonize the gut in two waves. (**a**) Enteric ganglia form the myenteric plexus (MP) between the circular (CM) and longitudinal muscle (LM) layers and the submucosal plexus (SMP) in the mucosa. (**b**) The majority of neural precursors are derived from the vagal neural crest, adjacent to somites 1 to 7 (orange). They migrate along the bowel in an oral to anal direction. A second group originates from sacral neural crest cells (in mice behind somite 24), forms the pelvic ganglia (pg) and migrates in an anal to oral direction (red). Further enteric neural precursors are descendants of Schwann cell precursors (SCP, green) or pancreatic progenitors (blue). (**c**) Neural-fated enteric neural crest-derived cells (ENCCs) (N) differentiate in the myenteric region (grey). In parallel, neuron-glia bipotent (N-G) and glial-fated ENCCs (G) migrate into the mucosal area (yellow) and differentiate further into neurons and glial cells.; h heart; ld liver diverticulum, pd pancreatic diverticulum, c caecum.

**Figure 2 cells-10-00047-f002:**
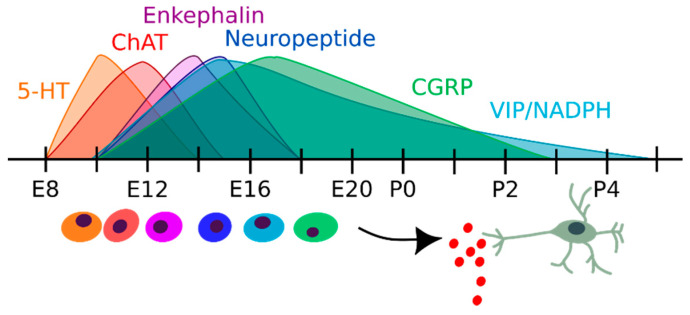
Time intervals in which different neuronal subtypes form. These are characterized by the expression of specific neurotransmitters.

**Figure 3 cells-10-00047-f003:**
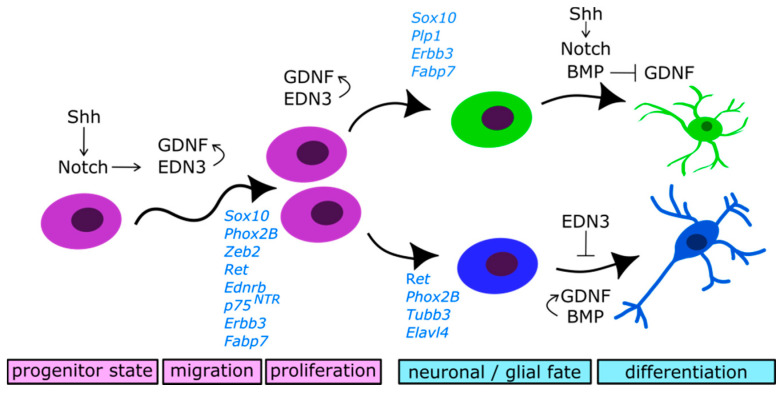
Schematic of enteric neural differentiation showing regulatory signalling pathways and marker gene expression. Several signalling pathways promote or inhibit key processes like ENCC (purple) migration, proliferation and differentiation of neuronal- and glial-fated progenitors (blue and green). Generally accepted markers for each cellular subpopulation are indicated in blue.

**Figure 4 cells-10-00047-f004:**
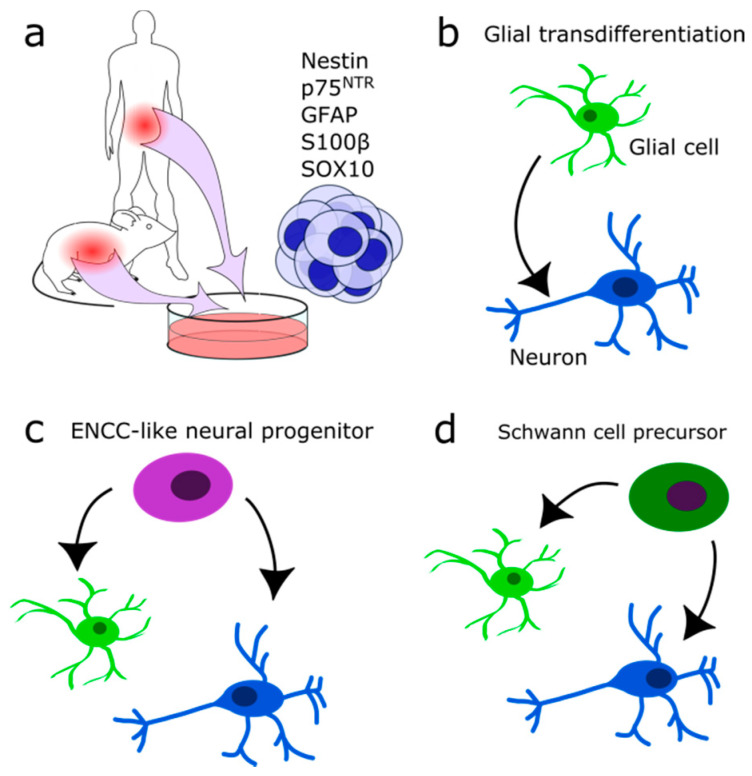
Suggested identities of multipotent neural progenitors. (**a**) Neurosphere-forming multipotent cells may be isolated from theintestinal system in human and mouse. (**b**) Glial cells may transdifferentiate into neurons directly after injury. (**c**) Adult ENCC-like neural progenitors with stem cell character may differentiate into neurons and glia during homeostasis and after injury. (**d**) Schwann cell precursor cells may differentiate into neurons and glial cells after injury.

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
