# Peer review of "Neuron–Glia Interaction in the Developing and Adult Enteric Nervous System"

_cells, 2020, doi:10.3390/cells10010047_

Round 1
Reviewer 1 Report
In this elegant review the authors addressed a very interesting topic that was investigated to some extent in recent years. Although it represents the largest part of the PNS the enteric nervous system il poorly investigated, even less the neuron-glial interaction, at the base of its development, maturation, regeneration under different physio-pathologic conditions. The review is updated, collecting most of the recent papers in the field. The ms. structure is very good, well-written, with consistent chapters. Interestingly, the ENS origin of neurons from pancreatic cells is fascinating and should be emphasized.
Some points would improve the readability and must be amended:
1) Line 108 SCP already abbreviated at line 76
2) Characterization of different neurotransmitter-expressing neurons is fundamental in the ENS. This has been reported at lines 158-167. References must be included. Moreover, a scheme, picture or figure summarizing those concepts would implement the paper.
3) It is generally accepted that genes are in italic, proteins in capital. Maybe this guideline was not always adopted. Please check throughout the text. For instance, gfap at line 201 is correct? Again line 220 GDNF/RET or EDN3/EDNRB in capitals is correct? EDN3/EDNRB abbreviated at line 242
4) Hedgehog significance, upstream notch, is emerging. This concept should be stressed at lines 284-on
5) Reference style line 341 must be adapted to the style
6) Regarding chapter 4, no mention on the role of microbioma and ENS progenitor alterations. Some paragraphs regarding this emerging correlation could be included.
Author Response
Reviewer 1
Interestingly, the ENS origin of neurons from pancreatic cells is fascinating and should be emphasized.
We are delighted that the reviewer finds the pancreatic origin of neurons as exciting as we do. To our knowledge, the cited articles are the firsts, which show this enteric-pancreatic interaction. We included the following sentence in line 106 to emphasize the content of the previous sentences in the text: “In total, both the gut and the pancreas supply each other with neural progenitors and thereby increasing the diversity of neuronal origin in the gastrointestinal system.”
1) Line 108 SCP already abbreviated at line 76
We excused the mistake and deleted the abbreviation in line 111.
2) Characterization of different neurotransmitter-expressing neurons is fundamental in the ENS. This has been reported at lines 158-167. References must be included. Moreover, a scheme, picture or figure summarizing those concepts would implement the paper.
Indeed, the arising of the different types of neurons deserves to be more accentuated. We introduced the corresponding reference in line 165 and implemented a new scheme illustrating the timing of neurotransmitter expression (Figure 2).
3) It is generally accepted that genes are in italic, proteins in capital. Maybe this guideline was not always adopted. Please check throughout the text. For instance, gfap at line 201 is correct? Again line 220 GDNF/RET or EDN3/EDNRB in capitals is correct? EDN3/EDNRB abbreviated at line 242
After rereading the manuscript, we concur that the sentence in line 228 might be misleading in this sense. We tried to save the word “expression” for genes and therefore meant to refer to the Gfap gene. To be more precise in the manuscript we changed the sentence into: ”Glial cells which produce GFAP at one time point might not exhibit it seven days later.”
The vast majority of publications uses the abbreviation in capitals for GDNF, RET and EDN3 and EDNRB. We followed this to be consecutive.
Indeed, the reviewer is right the abbreviation EDN3/EDNRB was used beforehand. We corrected this mistake and introduced the abbreviation now in line 260.
4) Hedgehog significance, upstream notch, is emerging. This concept should be stressed at lines 284-on
We assented to this opinion and tried to emphasise the Hh – Notch hierarchy linguistically with a new reference to Figure 3 (line 326) and small word implements, were inserted in line 327-334: “Further, Shh signalling may regulate Notch indirectly as well by modulating the composition of the ECM by upregulation of the synthesis of collagen I, collagen IX, chondroitin sulfate proteoglycan (GSPG) and versican [122]. ……. Hypothetically, Notch signaling in ENCCs may be regulated by the composition of the ECM, or mesenchymal cells may change ECM composition after Notch or Shh activation. Either way, the impact of Hh signaling via Notch on ENS development is very likely.”
5) Reference style line 341 must be adapted to the style
We thank the reviewer for drawing attention to this error. The reference was adapted to the formatting style.
6) Regarding chapter 4, no mention on the role of microbioma and ENS progenitor alterations. Some paragraphs regarding this emerging correlation could be included
The microbiome certainly is of significant importance for ENS function. We included a small section about this topic in chapter 4, line 534-549. With this newly applied changes, we aimed to cover this topic but also tried not to prolong the already extended length of the manuscript that much. We kept it briefly and referred to the respective literature: “First promising results have been obtained by the application of GDNF to HSCR mouse models [167]. As a consequence, survival rates of diseased animals improved, and even neurogenesis initiated by Schwann cells and remaining sacral ENCCs occurred. Further, GDNF improved epithelial barrier function and inflammatory responses in mouse models for ulcerative colitis [168]. Mucosal glial cells secrete GDNF after activation of the toll-like receptor 2 by the gut microbiome [169,170]. Enterocytes are the second source of GDNF but can sense GDNF signalling via the RET receptor. An activation of GDNF/RET signaling facilitated the formation of tight junctions and induced proliferation of enterocytes [98,169]. Thereby, GDNF supports homeostasis and wound healing of the gut epithelium.
Interactions of the gut microbiome and the ENS via the immune system is mandatory for normal functions of intrinsic and extrinsic nerves and gut-brain communication in homeostasis and allostasis [55,87,170–173]. Direct bacterial-neuronal interaction was mediated by 5-HT, calbindin and NO [9,16,171–173]. In turn, macrophages and mast cells interacted with enteric neurons via CSF and BMP 2 [87]. Novel drug treatments in combination with the restoration of the microflora or cell transplantation may lead to therapeutic success in a variety of neuronal and inflammatory diseases of the gastrointestinal tract.”
Reviewer 2 Report
This review paper targets two topics: review of the neuron-glia interaction in enteric nervous system (ENS) and the molecular biology of ENS development. The title of the paper covers only the first topic. The title does not cover the chapters about ontogeny (line 61-117). There are several reviews about ENS development, therefore the above listed chapters can be omitted or included in the title.
Although it is a well written manuscript, the previous publication from others significantly diminishes the novelty of this work. Further, no new insight on neuron-glia-epithelium or neuron-glia-macrophages (intraganglionic macrophages or ganglion associated macrophages), or glia+ other ENC accessory cells (Cajal cells, telocytes) integration process has been mentioned or delineated.
Figure 1: serosa layer is missing in Fig 1a,c
Figure 1: ceca as important structure for ENS is not illustrated in Fig 1C
Line 168: Drokhlyansky et al., 2020 identified 14 subsets of human and 18 subsets from the mouse enteric neuron in contrast to 9 and 7 in small and large intestine, respectively
Author Response
Reviewer 2
The title of the paper covers only the first topic. The title does not cover the chapters about ontogeny (line 61-117). There are several reviews about ENS development, therefore the above listed chapters can be omitted or included in the title.
We thank the reviewer for noticing this discrepancy. We now changed the title into:” Neuron-Glia interaction in the developing and adult enteric nervous system” for broader coverage of the content of this manuscript.
Further, no new insight on neuron-glia-epithelium or neuron-glia-macrophages (intraganglionic macrophages or ganglion associated macrophages), or glia+ other ENC accessory cells (Cajal cells, telocytes) integration process has been mentioned or delineated.
In this manuscript, we focussed on neuron-glia interaction as strictly as possible. However, we agree with the reviewer that other cell types like interstitial cells and macrophages are of enormous importance to enteric functions. Therefore, following the reviewers' suggestions we included from line 237-248, a brief overview of glia-macrophage interactions during development and in line 543-549 neuronal-macrophage interaction in homeostasis and allostasis. Further, from line 189-206, we introduced a new paragraph about interstitial cells of Cajal and telocytes and their role in signal transduction. Due to the already extended length of the manuscript, we pointed out the major aspects and referred to available literature of others for further reading.
Macrophages during development:
“The gut microbiota controls glial development and homeostatic renewal throughout the adult life. Consequently, antibiotic treatments impaired glial homeostasis [77,78]. The exact mechanism remains enigmatic, but one factor that links the microflora and the ENS are macrophages. Macrophages have a haematopoetic origin and colonise the embryonal gut independently of the ENS [79]. Distinct types of macrophages exhibit different transcriptomes, linked to specific functions and locations [80]. Mucosal macrophages modulated gut homeostasis and secretion by the interaction with neurons and blood vessels. In contrast, myenteric macrophages influenced ENS formation directly via the secretion of BMP (bone morphogenic proteins) molecules [81,82]. In turn, enteric glial cells activated intraganglionic macrophages via connexin 43 channels and the secretion of macrophage colony-stimulating factor (CSF) [83,84]. Further, they regulated group 3 innate lymphoid cells and thereby orchestrated gut defence [10]. Commonly, enteric glial cells are part of a tight regulatory circuit between the microbiota and the immune system [3,85–87].”
Macrophages in the adult gut:
“Interactions of the gut microbiome and the ENS via the immune system is mandatory for normal functions of intrinsic and extrinsic nerves and gut-brain communication in homeostasis and allostasis [55,87,170–173]. Direct bacterial-neuronal interaction was mediated by 5-HT, calbindin and NO [9,16,171–173]. In turn, macrophages and mast cells interacted with enteric neurons via CSF and BMP 2 [87]. Novel drug treatments in combination with the restoration of the microflora or cell transplantation may lead to therapeutic success in a variety of neuronal and inflammatory diseases of the gastrointestinal tract.”
Interstitial cells of Cajal and telocytes:
“An additional layer of signal control is mediated by interstitial cells of Cajal (ICC) and telocytes (TC), also referred to as fibroblast-like cells positive for the platelet-derived growth factor receptor α (PDGFRα+ cells) [3,56–58]. ICCs and TCs have a mesenchymal origin and TCs could differentiate into ICC upon damage [58–60]. Both cell types are found in several organs, including the gut, where they formed a network between and within the muscular layers. TCs are characterized by long telopods which are in contact with blood vessels, smooth muscle cells (SMC), nerves, macrophages and other immune cells [57,61,62]. TCs are believed to give mechanical support on a cellular level and facilitate cell-cell communication over cell contacts. In the intestinal crypts, subepithelial TCs support epithelial renewal and stem cell proliferation by secreting Wnt proteins. Additionally, they regulate epithelial gene expression in intestinal villi [63–65]. ICCs modulate electrical signal transduction from motorneurons to SMCs by generating a slow electrical wave which transduces phasic contraction of SMCs [3,56,57]. Cellular contact and gap junctions were identified between TCs and ICCs as well as ICCs and SMCs [58]. TCs might act as postjunctional cells involved in purinergic neurotransmission [62]. In the myenteric ganglia of the ileum, TCs formed a continuous layer around ganglia, whereas the ICCs were individually located between TCs and circular SMCs [66]. On the contrary, telopods were found inside ganglia in the colon. The functional coupling of SMCs, ICCs and TCs is necessary for organized peristaltic movements and defined them as a functional “SIP syncytium” (naming after SMCs, ICCs and PDGFRα+ cells) [56].”
Figure 1: serosa layer is missing in Fig 1a,c
Figure 1: ceca as important structure for ENS is not illustrated in Fig 1C
We are grateful for the close observation of the figures, and the resulting comment. The suggested changes were implemented in figure 1.
Line 168: Drokhlyansky et al., 2020 identified 14 subsets of human and 18 subsets from the mouse enteric neuron in contrast to 9 and 7 in small and large intestine, respectively
We sincerely apologize for this mistake. Zeisel and colleagues identified nine neural groups but could only include cells of the myenteric plexus of the small intestine in their analysis. The limitation was necessary due to the need of isolating neural cells from the tissue by FACS sorting in a Wnt1-Cre;R26Tomato transgenic line. Drokhlyansky and colleagues focused on the colon of mice and humans and were able to include all neural cells in the analysis. To be more evident in the text we changed the sentence into: “Two single-cell transcription analyses identified nine molecularly distinct enteric neuronal subsets in the myenteric plexus of the small intestine, 24 subgroups in the colon of mice and 11 in humans [53,54].” (line 175-177)
Reviewer 3 Report
This review by Pawolski and Schmidt nicely summarizes and discusses previous and recent studies regarding interactions between neuron and glial cells in the enteric nervous system in developmental, healthy and disease states. Overall, the review is well written and will be of interest to the field broadly. I enjoy reading the article. I have no major comments, but it would be nice to include recent information about tissue engineering of the enteric nervous system using human pluripotent stem cells (e.g., PMID: 27869805).
Author Response
Reviewer 3
to include recent information about tissue engineering of the enteric nervous system using human pluripotent stem cells (e.g., PMID: 27869805).
We thank the reviewer for commenting on the manuscript.
Indeed, techniques to generate intestinal organoids from human pluripotent stem cells advanced in the last years. In the suggested paper, the authors demonstrate that organoids initially restricted to the epithelium can be generated with a functional enteric neural network. We agree that a section about organoids fits into the topic and improves the manuscript. A paragraph about organoid formation is now present in line 511-517: “Advances in the generation of human intestinal organoids and enteroids may facilitate novel research on gastrointestinal diseases [160–162]. Initially, organoids were limited to epithelial structures but in combination with NCC derived from human pluripotent stem cells, organoids with a functional enteric neural network can be formed, enabling research on the ENS and associated diseases [163,164]. This opens new possibilities to test neural interactions and allows easy genetical and chemical manipulations. Organoids can be studied in culture or grafted into hosts which further amplifies possible scientific questions and potential therapeutics.”
Round 2
Reviewer 1 Report
The manuscript has been amended as suggested.
Reviewer 2 Report
-